# Clinical Impact of the Geriatric Nutritional Risk Index on Chemotherapy-Related Adverse Events in Diffuse Large B-Cell Lymphoma: A Multicenter Study

**DOI:** 10.3390/nu17233785

**Published:** 2025-12-02

**Authors:** Kei Fujita, Hikaru Tsukasaki, Shin Lee, Tetsuji Morishita, Eiju Negoro, Kana Oiwa, Takeshi Hara, Hisashi Tsurumi, Takanori Ueda, Takahiro Yamauchi

**Affiliations:** 1Department of Hematology and Oncology, University of Fukui, 23-3, Matsuoka-Shitago, Eiheiji-cho, Yoshida-gun, Fukui 910-1193, Japan; 2Department of Hematology and Oncology, Matsunami General Hospital, Dendai 185-1 Kasamatsu-cho, Hashima-gun, Gifu 501-6062, Japan; 3Department of Hematology, Juko-Osu Hospital, 2-2-43 Osu, Naka-ku, Nagoya 460-0017, Japan; 4Department of Hematology, Fukui Red Cross Hospital, 2-4-1 Tsukimi, Fukui 918-8501, Japan; 5Department of Internal Medicine, Matsunami General Hospital, Gifu 501-6062, Japan; 6Matsunami Research Park, Dendai 185-1 Kasamatsu-cho, Hashima-gun, Gifu 501-6062, Japan

**Keywords:** diffuse large B-cell lymphoma, Geriatric Nutritional Risk Index, nutritional assessment, toxicity prediction, real-world evidence

## Abstract

**Background/Objectives:** Accurate prediction of severe adverse events (SAEs) is crucial for optimizing supportive care while maintaining treatment intensity in diffuse large B-cell lymphoma (DLBCL). We evaluated the predictive value of the Geriatric Nutritional Risk Index (GNRI) for SAEs in de novo DLBCL and examined potential interactions with treatment regimen and age. **Methods:** This multicenter retrospective study included 555 adults treated with standard immunochemotherapies. SAEs, defined as grade ≥ 3 non-hematological adverse events or febrile neutropenia, were independently assessed by board-certified hematologists. **Results:** Multivariable logistic regression identified GNRI as an independent predictor of SAEs (odds ratio 0.982, 95% confidence interval 0.967–0.997). Restricted cubic spline modeling revealed a significant non-linear association between GNRI and SAE risk (*p* = 0.045). No significant interaction was observed between GNRI and regimen or age (*p* = 0.894 and 0.217, respectively), a finding consistent across subgroups in forest plot analyses. **Conclusions:** This study showed that lower diagnostic GNRI was independently associated with higher SAE risk regardless of treatment regimen or age. These findings highlight the potential utility of GNRI as a simple clinical indicator for identifying patients at higher risk of treatment-related toxicity, although they are derived from a retrospective, tertiary-care cohort and require confirmation in external prospective studies.

## 1. Introduction

Diffuse large B-cell lymphoma (DLBCL) is the most common subtype of non-Hodgkin lymphoma [1,2,3], and curative treatment depends on maintaining adequate relative dose intensity (RDI) of standard immunochemotherapy [4,5,6,7]. As treatment-related severe adverse events (SAEs) are a major cause of dose reductions or treatment discontinuations, accurate prediction of adverse event (AE) risk is crucial to optimizing supportive care and preserving dose intensity [8,9].

Malnutrition is a frequent but underrecognized comorbidity among patients with hematologic malignancies [10]. Impaired nutritional reserves and systemic inflammation may reduce hematopoietic recovery, delay bone marrow regeneration, and increase vulnerability to cytotoxic stress [11]. The Geriatric Nutritional Risk Index (GNRI) has been reported as a nutritional indicator that correlates with the prognosis of hospitalized elderly patients (age ≥ 65 years) [12]. Although low GNRI has been linked to inferior survival in DLBCL [13,14], its association with treatment-related toxicity has not been systematically investigated [15]. Further, although the predictive value of GNRI for AEs may differ depending on patient age or treatment regimen [16], only limited data are available on GNRI and chemotherapy tolerability in lymphoma [15], and prior studies have not systematically evaluated how these factors modify its predictive value for AEs.

To address this knowledge gap, the present study evaluated the usefulness of GNRI in predicting treatment-related SAEs, severe non-hematological toxicity, and febrile neutropenia (FN) in adult patients (age ≥ 18 years) with de novo DLBCL. We also examined whether these associations varied according to treatment regimen or patient age group.

## 2. Materials and Methods

### 2.1. Study Population and Clinical Information

This multicenter, retrospective analysis examined a cohort from three tertiary institutions in Japan: University of Fukui Hospital; the Japanese Red Cross Fukui Hospital; and Matsunami General Hospital. The medical records and oncology pharmacy records of consecutive patients diagnosed with de novo DLBCL during the period from 2006 to 2024 were reviewed. All data were collected by board-certified hematologists.

Lymphoma was diagnosed according to the World Health Organization classification [1,17]. Adult patients (age ≥ 18 years) with newly diagnosed de novo DLBCL who received at least one cycle of first-line standard immunochemotherapy with curative intent were included in this study [18,19,20]. We defined standard immunochemotherapy as R-CHOP (rituximab [RTX], cyclophosphamide [CPA], adriamycin [ADR], vincristine [VCR], and prednisolone [PSL]), R-THP-COP (RTX, CPA, tetrahydropyranyl adriamycin [THP], VCR, and PSL), and Pola-R-CHP (polatuzumab vedotin [Pola], RTX, CPA, ADR, and PSL) [18,19,20]. Patients were excluded if they had central nervous system involvement; composite lymphoma comprising DLBCL plus indolent lymphoma (transformed DLBCL); methotrexate-related DLBCL; post-transplant lymphoproliferative disorder; human immunodeficiency virus infection; were treated with palliative intent; received regimens other than R-CHOP, R-THP-COP, or Pola-R-CHP; or had missing baseline data required for GNRI calculation or SAE assessment.

Patient age, sex, anthropometric data (height and weight), performance status (PS), number of extranodal sites, Ann Arbor stage, elevated lactate dehydrogenase, serum albumin (Alb), International Prognostic Index (IPI), bulky mass (diameter > 7.5 cm), and B symptoms (fever, night sweats, or unintentional weight loss) were collected as baseline demographic data. Comorbidity and nutrition status at diagnosis were assessed using the Charlson Comorbidity Index (CCI) and GNRI [12,13,14,15,21,22].

### 2.2. Calculation of GNRI

The GNRI was calculated using a previously validated formula: GNRI = 14.89 × serum Alb (in grams per deciliter) + 41.7 × (actual body weight [in kilograms]/ideal body weight [in kilograms]). Ideal body weight was estimated as 22 × (height in meters)^2^, in accordance with standard anthropometric assumptions. For individuals for whom the actual body weight exceeded the ideal body weight, the weight ratio (actual/ideal) was capped at 1.0 to avoid overestimation of nutritional status. Patients were classified into four risk groups according to GNRI values: no risk, GNRI > 98; mild risk, GNRI 92–98; moderate risk, GNRI 82 to <92; and severe risk, GNRI < 82 [12].

Serum Alb levels used to calculate the GNRI were obtained from routine blood tests at diagnosis and were measured in the clinical laboratories of each participating hospital using automated chemistry analyzers, with regular internal and external quality control procedures performed according to institutional standards.

### 2.3. Definition and Calculation of RDI

The standard chemotherapeutic regimens in this study included R-CHOP, R-THP-COP, and Pola-R-CHP, all administered in a 21-day cycle. The R-CHOP regimen consisted of RTX at 375 mg/m^2^ intravenously on day 1, CPA at 750 mg/m^2^ intravenously on day 1, ADR at 50 mg/m^2^ intravenously on day 1, VCR at 1.4 mg/m^2^ (maximum 2.0 mg/body) intravenously on day 1, and PSL at 100 mg/body orally or intravenously on days 1–5 [18]. The R-THP-COP regimen was identical to R-CHOP in terms of dosing and schedule, except that THP at 50 mg/m^2^ replaced ADR [19]. The Pola-R-CHP regimen was a modified form of R-CHOP in which vincristine was replaced by Pola at 1.8 mg/kg [20].

Dose intensity (DI) was defined as the planned dose per unit time, calculated as the planned dose per cycle (mg/m^2^) divided by the planned duration of each cycle (weeks). RDI was expressed as the percentage of the delivered DI relative to the protocol-defined target DI. The average RDI (ARDI) was calculated as the mean RDI delivered per cycle for each chemotherapeutic agent (CPA, ADR or THP, and VCR or Pola). In this study, the RDI over all treatment cycles was defined as the total ARDI (tARDI). Receiving six standard cycles of standard regimens without any dose reduction or delay was defined as a tARDI of 100%. In patients receiving more than six cycles, tARDI could exceed 100%.

### 2.4. Outcome Measures

The primary outcome of this study was the occurrence of treatment-related SAEs, defined as Grade ≥ 3 non-hematological AEs and FN according to the Common Terminology Criteria for Adverse Events (CTCAE) version 5.0 from the National Cancer Institute [23]. AEs were evaluated using two approaches, and the primary analyses were based on a patient-based “overall assessment” of SAEs.

First, for an overall assessment across the entire treatment period, patients were considered to have experienced an SAE if they developed at least one SAE at any time during therapy. In this analysis, the number of events was equivalent to the number of patients who experienced at least one SAE. This overall assessment served as the primary outcome and as the dependent variable in all multivariable logistic regression and restricted cubic spline models.

Second, SAEs were also assessed on a per-cycle basis. If a patient experienced more than one SAE within a single treatment cycle, only the first event in that cycle was counted. When SAEs occurred across multiple treatment cycles, each cycle with an SAE was counted as a separate event. For irreversible AEs such as peripheral neuropathy, the event was counted only in the cycle in which it first appeared. This per-cycle assessment was used to descriptively characterize temporal patterns of toxicity and was not used as the primary outcome in regression modeling.

### 2.5. Statistical Analysis

Continuous variables are presented as median and range and were compared using the Mann–Whitney U test. Categorical variables are described as frequency and percentage and were analyzed using the chi-square test or Fisher’s exact test, as appropriate. Multivariable logistic regression analysis was performed to identify independent predictors of SAEs for an overall assessment across the entire treatment period. A nomogram was constructed based on the final multivariable logistic regression model to provide individualized absolute risk estimates of SAEs. Variables included in the model were sex, IPI, presence of bulky disease, CCI, tARDI, and GNRI. Restricted cubic spline (RCS) regression with three knots was used to assess potential non-linear associations between GNRI and incidence of SAEs in the primary analyses. Model calibration for the multivariable logistic regression was assessed using the Hosmer–Lemeshow goodness-of-fit chi-square test, and discriminative ability was summarized by the area under the ROC curve (AUC). Model calibration was also evaluated using bootstrap internal validation with 500 repetitions, generating optimism-corrected calibration slope, calibration intercept, and mean absolute calibration error. For the restricted cubic spline models, overall association and non-linearity of GNRI were evaluated using Wald chi-square statistics derived from the fitted models.

Receiver operating characteristic (ROC) curve analysis was conducted to evaluate the predictive performance of GNRI for SAEs. In addition, we pre-specified an exploratory analysis to derive a cohort-specific GNRI threshold for visualizing risk stratification. For this purpose, we used the ROC curve within this DLBCL cohort and identified the optimal cut-off based on the Youden index. This cohort-specific cut-off was used only to dichotomize GNRI in cycle-specific incidence plots, whereas all main inferences were based on GNRI as a continuous predictor in the spline models.

To assess the incremental predictive value of adding GNRI to established clinical variables, we compared a base logistic model including CCI alone with a model additionally incorporating GNRI. Model fit was compared using the Akaike Information Criterion (AIC), and improvement in model performance was tested using likelihood ratio statistics. Reclassification performance was evaluated using continuous Net Reclassification Improvement (NRI) and the Integrated Discrimination Improvement (IDI), calculated with nonparametric bootstrap procedures [24]. Clinical usefulness of the GNRI-augmented model was examined using decision curve analysis (DCA), in which the net benefit was plotted across threshold probabilities to compare the GNRI model against the base model [25].

Because the total sample size (*n* = 555) was modest, we did not use split-sample or temporal validation, in accordance with recent transparent reporting of a multivariable prediction model for individual prognosis or diagnosis plus artificial intelligence recommendations discouraging random or temporal data partitioning due to loss of statistical efficiency [26,27]. Instead, internal validity was assessed using bootstrap resampling (500 repetitions) with evaluation of optimism-corrected discrimination and calibration metrics. Subgroup analyses were conducted to examine interactions with age and treatment regimens, using adjusted logistic regression models incorporating RCS. Forest plots were generated to illustrate subgroup effects. All *p*-values were two-sided. *p*-values less than 0.05 and *p*-values for interactions less than 0.10 were considered significant. Statistical analyses were performed using R (version 4.2.1; The R Foundation for Statistical Computing, Vienna, Austria) or EZR (version 1.55; Saitama Medical Center, Jichi Medical University, Saitama, Japan) [28], as a graphical user interface for R [29].

## 3. Results

### 3.1. Patient Characteristics at Diagnosis and Details of SAEs

A total of 673 patients at the three participating tertiary institutions were pathologically confirmed to have DLBCL. Of these, 118 patients were excluded based on the exclusion criteria, and the remaining 555 patients were enrolled into this study (Appendix A). Patient characteristics at diagnosis of DLBCL are shown in Table 1. Median age at diagnosis was 74 years (range, 27–96 years), and 169 patients (30.5%) showed poor performance status (PS ≥ 2). A total of 342 patients (61.6%) experienced once or more SAEs throughout the entire treatment period. The most common treatment regimen was R-CHOP, administered to 451 patients (81.3%). R-THP-COP was given to 64 patients (11.5%), and Pola-R-CHP to 40 patients (7.2%). Regarding prophylactic measures, 69 patients (12.4%) received prophylactic oral antibiotics with levofloxacin, and 486 patients (87.6%) received prophylactic granulocyte colony stimulating factor (G-CSF) or pegylated G-CSF (PEG-G-CSF). The group without SAE showed significantly better GNRI than the group with SAE in terms of both continuous variables and categorical comparisons. On the other hand, no significant difference in tARDI was seen between groups with and without SAEs.

Table 2 shows the details of SAEs for each cycle of chemotherapy. FN was the most frequently observed SAE throughout the entire treatment period and across all treatment cycles. Despite the fact that approximately 90% of patients received primary prophylaxis with G-CSF or PEG-G-CSF, nearly half of the patients experienced at least one episode of FN during the course of treatment. Among non-hematological toxicities, anorexia and infections such as lung infection and SARS-CoV-2 disease were commonly observed, but incidences decreased over subsequent treatment cycles, similar to FN. In contrast, peripheral neuropathy, unlike other adverse events, emerged after the third cycle and gradually increased in frequency with successive cycles.

### 3.2. Relationship Between GNRI and Frequency of SAEs

The results of logistic regression analysis for clinical factors associated with the occurrence of SAEs are shown in Appendix A. In multivariable analysis, independent predictors of SAEs included the IPI (odds ratio [OR] 1.300, 95% confidence interval [CI] 1.120–1.510), CCI score (OR 1.180, 95% CI 1.030–1.340), and GNRI score (OR 0.982, 95% CI 0.967–0.997). No significant associations were identified between tARDI and occurrence of SAEs in either uni- or multivariable analyses. The overall fit of the multivariable logistic regression model was adequate, as indicated by a non-significant Hosmer–Lemeshow goodness-of-fit test (χ^2^ = 4.065, *p* = 0.851). Because tARDI may be influenced by early-cycle toxicities, we re-estimated the multivariable logistic regression model after removing tARDI to avoid potential post-treatment bias (Appendix A). In this sensitivity analysis, GNRI remained an independent predictor of SAE occurrence (OR = 0.983, 95% CI 0.968–0.998). IPI and CCI also demonstrated consistent effects compared with the original model. The exclusion of tARDI did not materially alter the overall conclusions, supporting the robustness of the association between lower GNRI and higher SAE risk.

The relationship between GNRI and frequency of SAEs was also evaluated in multivariable logistic modeling with RCS (Figure 1). The association between GNRI and risk of SAEs exhibited a non-linear pattern, as depicted by the RCS model (*p* for non-linearity = 0.045). As GNRI score increased, the log odds of SAE occurrence decreased steeply at lower GNRI values and the curve became markedly flatter around a GNRI of approximately 94–95, indicating that further improvements in nutritional status beyond this range were associated with only modest additional reductions in SAE risk.

According to the ROC analysis, the most discriminative cutoff value of the GNRI for SAE occurrence was 94 (sensitivity 57.6%, specificity 66.2%), with an area under the curve value of 0.630 (95% CI 0.583–0.677), which corresponds to a moderate level of discriminative ability (Appendix A). This ROC-derived threshold was broadly consistent with the flattening of the spline curve around GNRI 94–95. To evaluate model calibration, we performed bootstrap internal validation with 500 repetitions. The optimism-corrected calibration slope was 0.92 and the calibration intercept was −0.04. The mean absolute calibration error was 0.015, and the bias-corrected calibration curve closely approximated the ideal 45-degree line, indicating excellent agreement between predicted and observed SAE probabilities (Appendix A). To provide a clinically intuitive visualization of risk over treatment cycles, we therefore used GNRI < 94 vs. GNRI ≥ 94 as an exploratory stratification in the cycle-specific incidence plots (Figure 2), while all primary inferences regarding GNRI and SAE risk were based on treating GNRI as a continuous predictor in the spline models. SAEs were most frequent in the first treatment cycle for both high and low GNRI groups. The higher proportion of SAEs in the low GNRI group compared to the high GNRI group was particularly pronounced during the initial phase of treatment, up to cycle 5.

A bootstrap internal validation with 500 repetitions was performed using the rms package. The optimism-corrected Somers’ Dxy was 0.30 (corresponding to an AUC of approximately 0.65). The optimism-corrected calibration slope was 0.92, and the maximum calibration error (Emax) was 0.027, indicating good internal calibration and minimal overfitting.

To evaluate whether GNRI provides additional predictive information beyond established clinical factors, we compared a base logistic model including CCI alone with a model additionally incorporating GNRI. Adding GNRI significantly improved model fit, reducing the AIC from 640.5 to 634.8 (ΔAIC = −5.7). A likelihood ratio test confirmed that the GNRI-augmented model provided statistically significant improvement over the base model (χ^2^ = 4.72, *df* = 1, *p* = 0.030). Reclassification analyses further supported the incremental value of GNRI. The continuous NRI showed a modest, non-significant trend toward improved classification (NRI = 0.08; 95% CI −0.09 to 0.24). In contrast, the IDI demonstrated a small but statistically significant improvement in discrimination (IDI = 0.008; 95% CI 0.0004–0.0144). DCA indicated that the GNRI-augmented model provided consistently higher net benefit than the CCI-only model across clinically relevant threshold probabilities (approximately 10–45%), supporting the clinical utility of incorporating GNRI into baseline toxicity risk prediction (Appendix A).

### 3.3. Impact of Treatment Regimen and Age on Associations Between GNRI and SAEs

To assess whether treatment regimen had an interactive effect with the impact of GNRI on the occurrence of SAEs, we developed an RCS-logistic regression model for each regimen group (R-CHOP, R-THP-COP, and Pola-R-CHP) (Figure 3). Trajectories of the predicted risk of SAE across the GNRI score range demonstrated a highly consistent pattern among the three treatment regimens. This visual observation was further statistically corroborated by a non-significant interaction term for the treatment regimen (*p* for interaction = 0.894), suggesting that the relationship between GNRI and risk of SAEs was not substantially modified by the treatment regimen.

Similarly, we constructed an RCS logistic regression model to analyze the two age groups (<65 years vs. ≥65 years) (Appendix A). The curves illustrating the predicted risk of SAEs across the range of GNRI scores also showed a largely similar pattern for both age groups. This visual consistency was likewise statistically confirmed by a non-significant interaction term for age groups (*p* for interaction = 0.217), indicating no substantial modification of the GNRI-SAE relationship by age.

In addition to the RCS analysis, we further explored the potential interaction effects of treatment regimen and age on the relationship between GNRI and risk of SAEs using logistic regression modeling, presented as forest plots (Appendix A). The forest plot visually confirmed that ORs for the effect of GNRI on SAEs were highly consistent across all treatment regimens (R-CHOP, R-THP-COP, and Pola-R-CHP) and both age groups (<65 years vs. ≥65 years). Specifically, estimated ORs for a one-unit increase in GNRI ranged from 0.96 to 0.97 across these subgroups, with 95% CIs largely overlapping and showing no significant deviation from one another. No interaction effect of either age (*p* for interaction = 0.217) or treatment regimen (*p* for interaction = 0.894) was observed on the relationship between GNRI and SAE occurrence.

To facilitate individualized clinical application, we developed a nomogram based on the multivariable logistic regression model ( Appendix A). This nomogram integrates GNRI, IPI, and CCI to estimate absolute SAE risk at treatment initiation. The model demonstrated moderate discriminative ability (C-index = 0.67), supporting its utility for personalized toxicity risk estimation.

## 4. Discussion

This study identified a significant but distinctly non-linear association between GNRI at diagnosis and risk of SAEs in patients with de novo DLBCL receiving standard treatment regimens. The incidence of SAEs was highest during the first treatment cycle and progressively decreased in subsequent cycles. Further, the predictive utility of GNRI for SAEs was particularly pronounced during the early phase of treatment, especially within the initial cycles. Notably, our analysis revealed that a lower GNRI consistently increased the risk of SAEs, regardless of the treatment regimen or patient age, highlighting the robust and pervasive impact of nutritional status on treatment tolerability in this patient population.

A key strength of the present study is that it clarifies the unique contribution of GNRI to predicting chemotherapy-related toxicity in DLBCL, extending and refining the findings of prior GNRI research. A previous study evaluated GNRI in patients with heterogeneous lymphoma subtypes receiving R-CHOP, reporting an association between low GNRI and higher rates of adverse events and shorter treatment duration; however, that analysis was limited by its modest sample size, inclusion of mixed histologies, absence of non-linear modeling, and reliance on dichotomized GNRI categories [15]. In contrast, our study focused exclusively on de novo DLBCL, used SAEs as the primary toxicity endpoint, applied RCS modeling to evaluate GNRI as a continuous predictor, and incorporated multivariable adjustment that accounted for comorbidity and disease severity. This allowed us to demonstrate a distinctly non-linear dose–response relationship, with steeply increasing SAE risk at lower GNRI values and a plateau around GNRI 94–95—patterns that were not captured in previous work. Together, these features represent a meaningful advancement over earlier GNRI–toxicity studies by providing a more granular, DLBCL-specific and clinically applicable characterization of how nutritional vulnerability influences chemotherapy tolerability.

Our findings should also be interpreted in the context of current international recommendations on nutritional care in oncology. The European Society for Clinical Nutrition and Metabolism (ESPEN) guideline and practical guideline on nutrition in cancer patients emphasize that all patients with cancer should undergo routine screening for malnutrition and, if at risk, a more detailed nutritional assessment using validated tools [30]. In the hemato-oncology field, several observational studies and recent systematic reviews have shown that composite nutritional indices such as the Controlling Nutritional Status (CONUT) score are significantly associated with survival in patients with hematologic malignancies [31]. In DLBCL specifically, GNRI, Prognostic Nutritional Index (PNI), and CONUT have been reported as independent prognostic markers for overall survival and progression-free survival [32]. However, most of these studies have focused on long-term oncologic outcomes, and few have evaluated treatment-related toxicity as a primary endpoint. By demonstrating that baseline GNRI is strongly associated with the risk of severe adverse events during immunochemotherapy, our study extends this body of evidence and suggests that GNRI-based nutritional risk stratification may not only inform prognostication but also help identify patients who require more intensive supportive care during the early cycles of treatment.

In addition to GNRI, other composite nutritional indices such as the PNI and CONUT have been widely investigated in DLBCL. Comparative studies have shown that all three indices possess prognostic relevance; however, GNRI often demonstrates equal or superior predictive performance, particularly in newly diagnosed DLBCL cohorts, likely because it combines serum albumin with a weight-based indicator of chronic nutritional decline rather than relying solely on lymphocyte counts or total cholesterol [16,32]. Given that early-cycle treatment-related toxicity is strongly influenced by host frailty and systemic inflammation, GNRI may more appropriately capture this vulnerability than PNI or CONUT, which are more sensitive to transient inflammatory changes. Nevertheless, our study did not directly compare GNRI with these indices or evaluate the independent contributions of its components. Future studies incorporating head-to-head comparisons of GNRI, PNI, and CONUT in toxicity-focused models will be essential to clarify their relative clinical utility.

Our findings highlight a crucial aspect of nutritional status in DLBCL patients: a lower GNRI consistently exerted a negative impact on the risk of SAEs, irrespective of age or treatment regimen. The GNRI has been investigated for its prognostic utility across various age groups of DLBCL patients [13]. More recently, we demonstrated that, irrespective of age, patients are equally affected by the negative prognostic impact of malnutrition [16]. Confirming earlier studies, our study also observed that a lower GNRI was associated with an increased risk of SAEs irrespective of age. Similarly to the analysis for age, the negative impact of a low GNRI on SAE risk was uniformly observed across all regimens. The robustness of this trend is underscored by its consistent observation across two distinct analytical approaches: RCS models, and forest plots. Although we could not directly compare GNRI with other inflammation- or nutrition-based indices, such as PNI or CONUT, because baseline lymphocyte count was not consistently available, prior work has already established GNRI as an independent prognostic marker in DLBCL [32]. Our study further expands its clinical relevance by demonstrating its association with treatment-related toxicity.

Another major strength of this study was the use of meticulously curated clinical data to evaluate AEs, which are inherently variable and often difficult to assess consistently in real-world settings. Accurate AE reporting is known to be challenging in clinical trials [9], and prior studies have documented frequent underreporting when standard methods are used [33,34]. One key contributor to underreporting is the inadequate understanding of AEs and study protocols among data collectors. In this study, all clinical data were collected and reviewed by board-certified hematologists, ensuring consistency in expertise and strict protocol adherence in toxicity assessment based on CTCAE. As a result, consistency in data quality and interpretation was maintained across participating institutions. Grade ≥ 3 non-hematological AEs and FN were most frequently observed during the first treatment cycle, with incidences decreasing by nearly half in subsequent cycles. These findings suggest that for patients with low GNRI at diagnosis, inpatient care and close monitoring during the initial cycle may be warranted to reduce the risk of SAEs and maintain dose intensity.

An additional clinically relevant finding of our study is the high incidence of FN despite widespread use of growth factor prophylaxis. Nearly half of the patients (49.9%) experienced at least one episode of FN, even though 87.6% received primary prophylaxis with G-CSF or PEG-G-CSF. This may partly reflect the frailty of our real-world cohort, in which the median age was 74 years (range, 27–96 years) and 30.5% of patients had an ECOG PS ≥ 2, contrasting with the generally younger and fitter populations enrolled in randomized clinical trials [18,20]. This pattern suggests that standard supportive care may be insufficient to offset the vulnerability conferred by poor baseline host factors, such as malnutrition and a high comorbidity burden. In our multivariable models, both GNRI and CCI independently predicted the overall occurrence of SAEs, supporting the concept that patients with impaired nutritional status and multiple comorbidities constitute a residual high-risk group in whom conventional growth factor support alone may not be adequate. In such patients, GNRI-based risk stratification could help to identify candidates for intensified infection surveillance, early nutritional intervention, or more cautious dose planning in addition to routine G-CSF prophylaxis. However, because our study did not standardize the timing, dosing, or formulation of G-CSF across centers, we cannot exclude the possibility that heterogeneity in prophylaxis practices also contributed to the observed FN rates. Additionally, formal calibration assessment using bootstrap resampling demonstrated excellent agreement between predicted and observed risks (optimism-corrected slope 0.92; MAE 0.015), supporting the model’s reliability and clinical applicability. When interpreting the cycle-specific decrease in SAE incidence, however, it is important to recognize the potential for survivor bias; patients who experience early toxicity or early treatment modification—particularly those with low GNRI—may contribute fewer person-cycles at risk later in treatment, leading to an underestimation of late-cycle SAE rates.

In defining the SAE endpoint, we deliberately focused on FN and grade ≥ 3 non-hematologic toxicities that typically require unplanned hospitalization, intravenous antibiotics or substantial dose modifications. FN in particular has been repeatedly reported as a major driver of RDI reduction and unplanned admissions during CHOP- or R-CHOP-based therapy [35]. Although asymptomatic grade 3–4 hematologic toxicities, such as neutropenia without fever, can also lead to treatment delays or dose reductions in some situations, these events are generally anticipated consequences of myelosuppressive chemotherapy and are often managed by scheduled monitoring, G-CSF support and short-term adjustments in cycle timing. Accordingly, our SAE definition prioritized clinically overt complications that directly affect treatment continuity and patient experience. We acknowledge, however, that this approach may have missed a subset of severe hematologic events without fever that still influenced dose intensity, and this should be considered when interpreting our findings.

GNRI emerged as a robust predictor of severe adverse events in patients with DLBCL, and several complementary analyses demonstrated its incremental value beyond conventional clinical variables. The addition of GNRI significantly improved overall model fit and likelihood ratio statistics, and also resulted in a small but statistically significant increase in IDI, indicating improved discrimination. Although the continuous NRI did not show statistically significant reclassification improvement, the direction of effect was consistent with the observed gains in discrimination. Importantly, the DCA revealed that the GNRI-augmented model achieved consistently higher net benefit than the CCI-only model across a wide range of clinically relevant threshold probabilities. This finding indicates that the predictive contribution of GNRI is not merely statistical but also translates into meaningful clinical utility, potentially supporting more informed toxicity risk stratification at treatment initiation. While GNRI alone cannot replace comprehensive nutritional and functional assessments, its simplicity, availability at baseline, and demonstrated predictive value position it as a practical complement to existing risk prediction tools. Incorporating GNRI may help clinicians identify patients who could benefit from closer monitoring, proactive supportive care, or treatment individualization during early chemotherapy cycles. Nevertheless, the discriminative performance of the model was only moderate (AUC 0.63), indicating that GNRI alone cannot provide strong individualized prediction. Rather, its utility may lie in serving as a simple pre-treatment screening tool that complements other clinical and inflammatory predictors. Although our study spanned a long period (2006–2024), core principles of DLBCL management have remained broadly stable in Japan. Moreover, the association between GNRI and SAE risk was highly consistent across treatment regimens, including Pola-R-CHP, indicating that temporal variability is unlikely to have materially influenced our findings.

A further noteworthy finding of our study is that tARDI was not significantly associated with the overall occurrence of SAEs. This finding should be interpreted with caution because tARDI is a post-baseline, treatment-dependent variable rather than a fixed pre-treatment characteristic. In routine clinical practice, dose intensity is frequently modified in response to emerging toxicities, particularly in patients with poor baseline nutritional status or a high comorbidity burden. In our previous multicentre study of elderly patients with DLBCL, both GNRI and CCI were significant determinants of whether patients received standard immunochemotherapy, indicating that physicians already adapt treatment intensity according to the patient’s nutritional and comorbidity profiles in real-world practice [14]. It is therefore plausible that patients with low GNRI and higher CCI scores in the present cohort experienced SAEs early in the course of therapy and subsequently underwent dose reductions or delays, which mitigated the risk of further SAEs in later cycles. In such situations, lower tARDI may reflect appropriate toxicity management tailored to the patient’s vulnerability rather than a higher intrinsic susceptibility to adverse events, thereby attenuating the apparent association between tARDI and the binary overall SAE outcome. Moreover, tARDI averages dose intensity across the entire treatment period and may not adequately capture transient high-intensity exposure during the first few cycles, which were the most critical for SAE occurrence in our cohort. Finally, patients who discontinued treatment early because of severe toxicity or disease progression inevitably had lower tARDI but less time at risk for later-cycle SAEs, introducing survivor bias and further weakening the relationship. Taken together, these considerations suggest that baseline host factors such as nutritional status and comorbidity burden may be more informative for pre-treatment SAE risk stratification than global dose-intensity metrics like tARDI.

Our study has several limitations. First, since the investigation was conducted at tertiary-care centers, a healthcare access bias may have been present as an inherent form of selection bias. In our cohort, the median age at diagnosis was 74 years and approximately one-third of patients had an ECOG PS ≥ 2, indicating that a substantial proportion of older and frail individuals were treated at these institutions. As a population-based reference, nationwide cancer registry data from Japan reported a median age at diagnosis of 73.2 years among 214,209 patients with malignant lymphoma between 2016 and 2021 [36]. Although these registry data are not specific to DLBCL, they indicate that our cohort is comparable in age to the general Japanese lymphoma population, and the high proportion of ECOG PS ≥ 2 in our study suggests that many of our patients were clinically frail. These findings thus may not be generalizable to patients in other healthcare settings or with limited access to specialized care. Second, the retrospective design introduced information bias. AE prevention strategies were not standardized but left to the discretion of individual primary physicians. Third, an important limitation relates to the use of serum Alb as a key component of the GNRI. Serum Alb is a negative acute-phase protein and can be influenced not only by chronic undernutrition but also by acute inflammatory responses, liver dysfunction, fluid status, and other comorbid conditions [37,38]. In this retrospective study, albumin levels were obtained from routine blood tests at diagnosis, and detailed information on concurrent inflammatory markers or transient infections was not systematically available. Although measurement quality was ensured by standardized automated chemistry analyzers with regular internal and external quality control, we could not fully disentangle the effects of malnutrition from those of inflammation or other non-nutritional factors on GNRI values. Consequently, GNRI in this setting should be interpreted as a composite indicator of overall vulnerability that reflects both nutritional and inflammatory status, rather than a purely nutritional measure. In addition, recent studies have proposed objective biochemical and hematologic markers for identifying undernutrition in older adults, further contextualizing the interpretation of Alb-based indices such as GNRI [39]. Moreover, dysregulated adipokine signaling and oxidative stress have been implicated in the pathophysiology of malnutrition [40], supporting the biological plausibility that impaired nutritional reserves may increase susceptibility to treatment-related toxicity. Fourth, the reduction in SAEs during later treatment cycles should be interpreted with caution due to potential survivor bias. This trend may reflect appropriate dose delays or reductions following early-cycle SAEs, allowing treatment continuation. Alternatively, it may indicate that only patients with sufficient physical resilience proceeded to later cycles. Finally, most patients in this study received R-CHOP, while the numbers treated with R-THP-COP or Pola-R-CHP were limited. In particular, Pola-R-CHP is a recently established standard regimen [20], and further accumulation of real-world safety data is warranted.

## 5. Conclusions

This study revealed a significant, non-linear association in which a lower GNRI consistently predicted a higher risk of SAEs among patients with de novo DLBCL who received standard regimens, irrespective of treatment regimen or age. These findings suggest that the GNRI may serve as a simple and readily available indicator to support identification of patients at higher risk of treatment-related toxicity and to assist in planning early supportive care; however, it should be used in conjunction with comprehensive nutritional assessment, and further prospective and validation studies are needed before GNRI can be incorporated into routine clinical guidelines.

## Figures and Tables

**Figure 1 nutrients-17-03785-f001:**
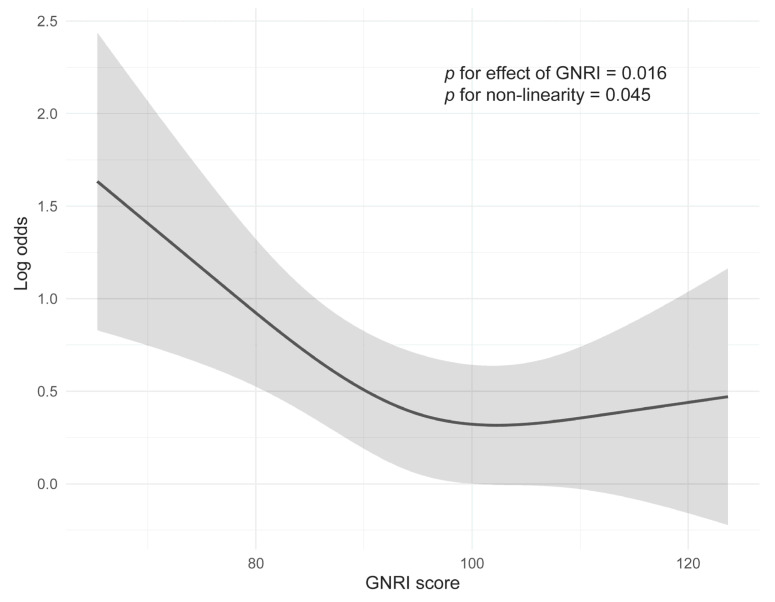
Covariable-adjusted logistic model with restricted cubic spline showing the association between GNRI and the risk of severe adverse events. The solid line represents the log odds ratio, and the shaded area is the 95% confidence interval. GNRI = geriatric nutritional risk index.

**Figure 2 nutrients-17-03785-f002:**
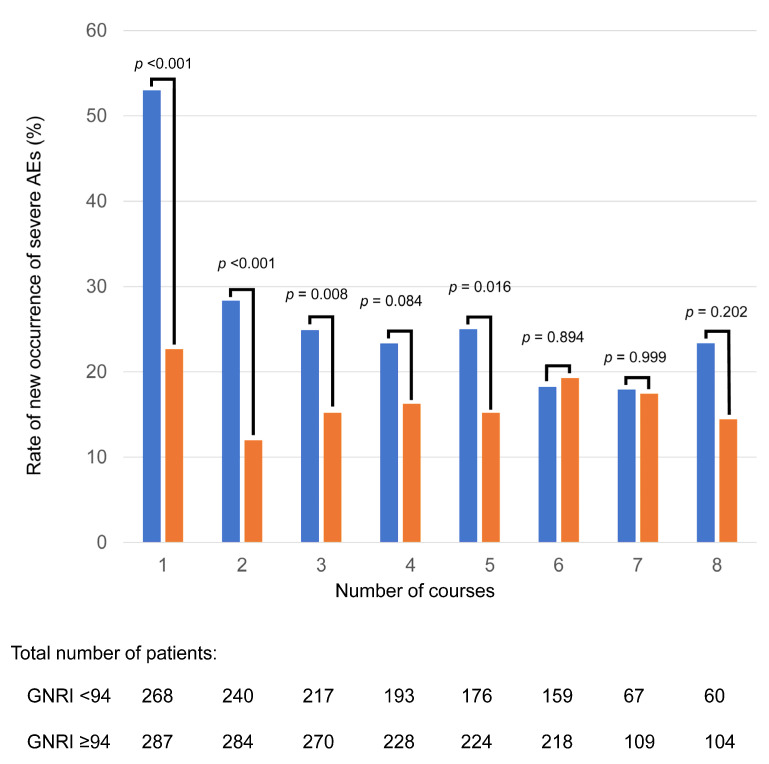
Bar graph showing the rate of new occurrences of severe adverse events in each treatment cycle binary group divided by the GNRI (<94 vs. ≥94). The blue bar represents the rate of new occurrence of severe adverse events in low GNRI group, and the orange bar represents the rate of new occurrence of severe adverse events in high GNRI group. GNRI = geriatric nutritional risk index.

**Figure 3 nutrients-17-03785-f003:**
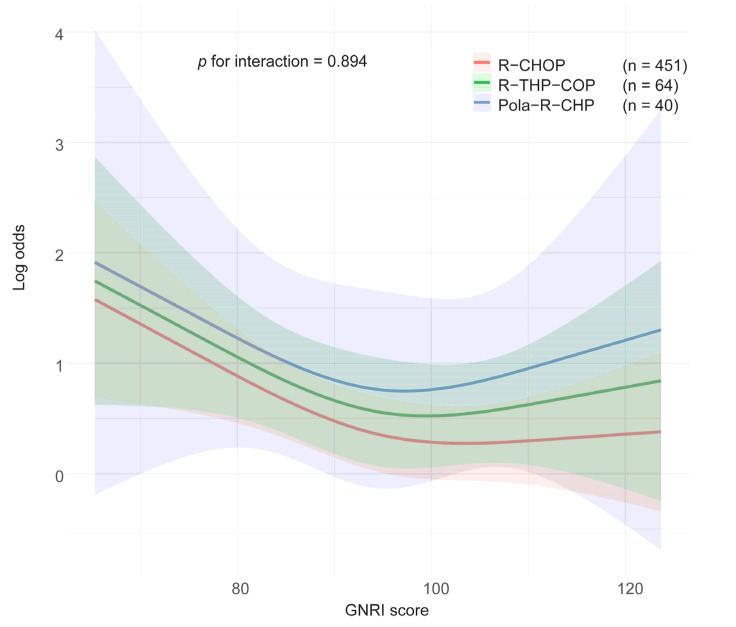
Covariable-adjusted logistic model with restricted cubic spline with showing the association between GNRI and the risk of severe adverse events according to the treatment regimens. The solid line represents the log hazard ratio, and the shaded area is the 95% confidence interval. GNRI = geriatric nutritional risk index; R-CHOP = rituximab, cyclophosphamide, adriamycin, vincristine, prednisolone; R-THP-COP = rituximab, cyclophosphamide, tetrahydropyranyl adriamycin, vincristine, prednisolone; Pola-R-CHP = polatuzumab.

**Table 1 nutrients-17-03785-t001:** Patients’ characteristics at diagnosis.

	All Patients (*N* = 555)		With Severe AEs (*n* = 342)		Without Severe AEs (*n* = 213)		*p*-Value
Age, years-median, range	74	(27–96)	75	(35–96)	72	(27–96)	0.008
Male-*n* (%)	284	(51.2)	170	(49.7)	114	(53.5)	0.384
ECOG PS ≥ 2-*n* (%)	169	(30.5)	134	(39.2)	35	(16.4)	<0.001
Extranodal sites ≥ 2-*n* (%)	205	(36.9)	154	(45.0)	51	(23.9)	<0.001
Ann Arbor Stage III/IV-*n* (%)	361	(65.0)	240	(70.2)	121	(56.8)	0.002
Elevated LDH (>ULN)-*n* (%)	350	(63.1)	230	(67.3)	120	(56.3)	0.011
Serum albumin (g/dL)-median, range	3.6	(1.1–5.6)	3.4	(1.1–5.6)	3.8	(1.4–5.0)	<0.001
IPI-*n* (%)							
Score-median, range	33	(0–5)	3	(0–5)	2	(0–5)	<0.001
Low (0, 1)	121	(21.8)	57	(16.7)	64	(30.1)	
Low intermediate (2)	117	(21.1)	62	(18.1)	55	(25.8)	<0.001
High intermediate (3)	112	(20.2)	67	(19.6)	45	(21.1)	
High (4, 5)	205	(36.9)	156	(45.6)	49	(23.0)	
Bulky mass-*n* (%)	132	(23.8)	91	(26.6)	41	(19.3)	0.058
B symptoms-*n* (%)	157	(28.3)	120	(35.1)	37	(17.4)	<0.001
CCI-*n* (%)							
Score-median, range	1	(0–8)	1	(0–8)	1	(0–6)	0.002
0	216	(38.9)	120	(35.1)	96	(45.1)	
1, 2	235	(42.3)	142	(41.5)	93	(43.7)	<0.001
3, 4	78	(14.1)	57	(16.7)	21	(9.9)	
≥5	26	(4.7)	23	(6.7)	3	(1.4)	
GNRI-*n* (%)							
Score-median, range	94.8	(38.3–136.0)	91.9	(38.3–129.2)	99.2	(60.5–136.0)	<0.001
No risk (>98)	227	(40.9)	113	(33.0)	114	(53.5)	
Mild (92–98)	91	(16.4)	54	(15.8)	37	(17.4)	<0.001
Moderate (82 to <92)	137	(24.7)	95	(27.8)	42	(19.7)	
Severe (<82)	100	(18.0)	80	(23.4)	20	(9.4)	
Total ARDI, –median, range	89.6	(4.9–176.6)	87.9	(4.9–176.6)	91.4	(8.2–143.3)	0.280
Prophylactic oral antibiotics-*n* (%)	69	(12.4)	48	(14.0)	21	(9.9)	0.194
Prophylactic G-CSF-*n* (%)	486	(87.6)	321	(93.9)	165	(77.5)	<0.001
Chemotherapy-*n* (%)							
R-CHOP	451	(81.3)	272	(79.5)	179	(84.0)	0.403
R-THP-COP	64	(11.5)	44	(12.9)	20	(9.4)	
Pola-R-CHP	40	(7.2)	26	(7.6)	14	(6.6)	

Data are presented as median (range) for continuous variables and number (percentage) for categorical variables. *p*-values were calculated using the Mann–Whitney U test for continuous variables and the chi-square test or Fisher’s exact test, as appropriate, for categorical variables. ARDI = average relative dose intensity. CCI = Charlson Comorbidity Index. ECOG PS = Eastern Cooperative Oncology Group performance status. G-CSF = granulocyte colony stimulating factor. GNRI = Geriatric Nutritional Risk Index. IPI = International Prognostic Index. LDH = Lactate dehydrogenase. ULN = Upper limit of normal. R-CHOP = rituximab, cyclophosphamide, adriamycin, vincristine, prednisolone; R-THP-COP = rituximab, cyclophosphamide, tetrahydropyranyl adriamycin, vincristine, prednisolone; Pola-R-CHP = polatuzumab vedotin, rituximab, cyclophosphamide, adriamycin, prednisolone.

**Table 2 nutrients-17-03785-t002:** Occurrence of severe non-hematological toxicity (grade ≥ 3) and febrile neutropenia.

Adverse Event	All Duration (*n* = 555) *n* (%)	Course 1 (*n* = 555) *n* (%)	Course 2 (*n* = 524) *n* (%)	Course 3 (*n* = 487) *n* (%)	Course 4 (*n* = 421) *n* (%)	Course 5 (*n* = 400) *n* (%)	Course 6 (*n* = 377) *n* (%)	Course 7 (*n* = 176) *n* (%)	Course 8 (*n* = 164) *n* (%)
Any non-hematological toxicity	200 (36.0)	94 (16.9)	45 (8.6)	44 (9.0)	33 (7.8)	27 (6.8)	23 (6.1)	15 (8.5)	11 (6.7)
Anorexia	33 (6.0)	9 (1.6)	5 (1.0)	6 (1.2)	4 (1.0)	2 (0.5)	3 (0.8)	2 (1.1)	2 (1.2)
Lung infection	32 (5.8)	9 (1.6)	10 (1.9)	3 (0.6)	4 (1.0)	5 (1.3)		1 (0.6)	
Sepsis	30 (5.4)	13 (2.3)	4 (0.8)	5 (1.0)	3 (0.7)	3 (0.8)			2 (1.2)
Peripheral neuropathy	20 (3.6)			2 (0.4)	3 (0.7)	4 (1.0)	6 (1.6)	3 (1.7)	2 (1.2)
Transaminase increased	18 (3.2)	8 (1.4)	2 (0.4)	2 (0.4)	4 (1.0)	2 (0.5)			
Hyperglycemia	15 (2.7)	3 (0.5)	2 (0.4)	2 (0.4)	2 (0.5)	3 (0.8)	2 (0.5)	1 (0.6)	
COVID-19 infection	10 (1.8)	3 (0.5)	2 (0.4)			2 (0.5)	3 (0.8)		
Tumor lysis syndrome	10 (1.8)	9 (1.6)	1 (0.2)						
Delirium	9 (1.6)	3 (0.5)	2 (0.4)	1 (0.2)		1 (0.3)	1 (0.3)		1 (0.6)
Ileus	9 (1.6)	4 (0.7)	2 (0.4)	1 (0.2)		2 (0.5)			
Febrile neutropenia	277 (49.9)	155 (27.9)	70 (13.4)	68 (14.0)	65 (15.4)	62 (15.5)	57 (15.1)	20 (11.4)	23 (14.0)
Treatment-related mortality	25 (4.5)	7 (1.3)	5 (1.0)	8 (1.6)	2 (0.5)	2 (0.5)	1 (0.3)		

COVID-19 = SARS-CoV-2 disease.

## Data Availability

The original contributions presented in this study are included in the article/Appendix A. Further inquiries can be directed to the corresponding author.

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
