# Peer review of "Clinical Impact of the Geriatric Nutritional Risk Index on Chemotherapy-Related Adverse Events in Diffuse Large B-Cell Lymphoma: A Multicenter Study"

_nutrients, 2025, doi:10.3390/nu17233785_

Round 1

Reviewer 1 Report

Comments and Suggestions for Authors

This study of 555 adult patients with diffuse large B-cell lymphoma (DLBCL) found that a lower Geriatric Nutritional Risk Index (GNRI) at diagnosis is a strong, independent predictor of severe adverse events (SAEs) during immunochemotherapy. The association is non-linear, with risk decreasing sharply until a GNRI of 94, and holds true regardless of age or treatment regimen (R-CHOP, R-THP-COP, or Pola-R-CHP). The study concludes that GNRI is a simple, effective tool for identifying patients at high risk of treatment-related toxicity, especially in early cycles.

Major strengths include a robust cohort size, meticulous data quality reviewed by hematologists, advanced statistical modeling (Restricted Cubic Spline) to reveal the non-linear relationship, and comprehensive subgroup analyses confirming the robustness of GNRI's predictive power.

Areas for clarification include:

  • GNRI Cutoff and Predictive Modeling: While the study uses GNRI as a continuous variable effectively, the use of an empirically derived cutoff of 94 for cycle-specific incidence charts is inconsistent with the primary finding of a non-linear relationship. A better justification for this cutoff or presentation using validated GNRI risk groups is recommended.
  • Definition of Primary Outcome: The text should explicitly state that the primary findings (ORs for IPI, CCI, and GNRI, and the RCS model) are based on the "Overall Assessment" of SAEs (patient-based outcome) to avoid ambiguity.
  • Discussion of Prophylaxis and Febrile Neutropenia (FN): The high incidence of FN (49.9%) despite extensive G-CSF prophylaxis (87.6% of patients) is a significant clinical observation that should be emphasized more strongly in the discussion. It suggests that malnutrition and high comorbidity may diminish the effectiveness of standard supportive care.
  • Interpretation of tARDI Findings: The lack of a significant association between total Average Relative Dose Intensity (tARDI) and SAEs warrants more discussion. The authors should speculate whether dose modifications acted as a mitigating factor for toxicity or if initial toxicity was too high for some patients.

Overall, the manuscript provides a robust and clinically relevant analysis, making valuable contributions to the literature by establishing GNRI as a critical and independent predictor of chemotherapy-related toxicity in DLBCL.

Reviewer 2 Report

Comments and Suggestions for Authors

Dear Authors,

Your study addresses a clinically significant question: how a simple assessment of nutritional status translates into treatment tolerance in DLBCL. The GNRI, which combines albumin and the body-weight–to-height relationship, is readily available and does not require specialized calculations. In a large, homogeneous cohort, you show that a lower GNRI is associated with a higher probability of severe AEs. This has immediate implications for planning supportive care and monitoring during the early cycles. The use of restricted cubic splines (RCS) captures the nonlinearity of the effect and suggests a practical threshold around 95, which is more informative than simple risk categories. At the same time, AUC=0.63 indicates moderate discriminative ability, implying that GNRI should likely serve as a component rather than the sole basis for decisions. It is valuable that the GNRI effect remains consistent across regimens (R-CHOP, R-THP-COP, Pola-R-CHP) and age groups, which enhances generalizability. An important observation is the high rate of FN despite widespread G-CSF prophylaxis, underscoring the need for risk stratification at baseline. From an interpretive standpoint, the early cycles seem most “critical,” which may justify intensified monitoring for patients with GNRI <95. Your adjudication of AEs by hematologists increases the credibility of toxicity classification. On the other hand, including tARDI in the SAE-risk model is methodologically questionable because tARDI depends on the very toxic events being modeled. In addition, the absence of a formal calibration assessment limits the clinical utility of the predictor. There is no evaluation of GNRI’s incremental value beyond classical variables (age, IPI, PS, CCI) using reclassification metrics. The SAE endpoint was restricted to non-hematologic events and FN; while defensible, this may omit clinically relevant severe hematologic AEs without fever. Variability in practice and calendar time (2006–2024, the COVID-19 era, Pola-R-CHP) may introduce heterogeneity. It would be worthwhile to compare GNRI with other indices (PNI, CONUT) and analyze its components. To broaden the context, please consider citing: DOI:10.3390/jcm14051494 — this article provides clinical and laboratory criteria for identifying undernutrition in older adults, complementing the interpretation of nutritional indices and albumin in onco-hematology; and DOI:10.3390/antiox12030569 — this paper links oxidative stress and adipokines with undernutrition, strengthening the biological rationale for nutritional parameters in predicting toxicity. It would also be helpful to present a nomogram or a table of absolute risk predictions for clinicians. In summary, the work is valuable, pragmatic, and close to clinical practice, but it requires several enhancements to realize the GNRI signal's clinical potential fully. Please respond point by point to the above suggestions and indicate which analyses/additions have been implemented and which are not feasible.

Best regards,

The reviewer.

Reviewer 3 Report

Comments and Suggestions for Authors

Dear author,

Thank you to give the opportunity to review your manuscript. I have some suggestions to improve it.

Title

Consider include “Multicentric study” instead of “A Real-World Study”.

Abstract

I recommend to include limitations and potential biases.

Methods

Define clearer the inclusion and exclusion criteria, making it easier to understand at a glance.

Ethics approval details and an IRB number are provided. But clarify waiver of written consent and how patient confidentiality/data protection were handled.

Mention laboratory quality control, calibration, or standard operating procedures for albumin measurement.  

Results

Table 1. Include the statistical method for stablishing the significant difference at the bottom.

No mention of logistic regression model checks (e.g., goodness-of-fit, ROC curve validation, or spline model diagnostics). These are necessary for reporting integrity.

Discussion

Misses international guidelines or systematic reviews on nutrition assessment in haematological malignancy and DLBCL.

Limitations

Recruitment from tertiary centers strongly limits generalizability. A more quantitative assessment, would be useful, for example “a demographic comparison to Japanese national registry”

Albumin levels can be affected by acute phase response, hydration, and laboratory variability. This should be discussed in greater depth.

Conclusion

Reword statements to avoid suggesting GNRI should replace comprehensive nutritional assessment or be implemented directly in guidelines without further research.

Reviewer 4 Report

Comments and Suggestions for Authors

While GNRI had been linked with prognosis in DLBCL before, the manuscript needs a stronger statement of what is actually novel here beyond validation of known information. The authors mention “no previous study examined toxicity prediction” several times, but must directly compare to prior GNRI–toxicity work (eg, Kikuchi et al. In Vivo 2023) rather than implying a total vacuum of prior knowledge. Please add a table or paragraph directly comparing.

The study only assesses GNRI in isolation, without any comparison to PNI, CONUT, NLR, GPS/mGPS, or LIPI all of which are also utilized in lymphoma toxicities. This is a severe methodological gap at minimum, this must be acknowledged with some justification and preferably with a concurrent multivariable model including competing nutrition indices.

The model’s AUC (receiver operating curve) of 0.630 is very weak. The manuscript’s current interpretation of GNRI as “strongly useful” is quite overstated. The authors should discuss that predictive accuracy is only moderate and either how it compares to dosage-based or inflammatory predictors or reframe the model as a potential pre-screening option only.

No validation cohort or internal resampling (bootstrapping/cross-validation). That severely limits real-world reliability. At minimum:

Consider split-sample or temporal validation if the data across 2006-2024 allows
The authors briefly note “possible survivor bias” but neither analyze nor adjust for it. Many low-GNRI patients may have been subject to early dose modification or early death, artificially lowering the observed SAE risk in later cycles. This must be analyzed or at least explicitly discussed as a confounder.

Round 2

Reviewer 3 Report

Comments and Suggestions for Authors

Dear Authors, 

Changes have been applied correctly. 

Best regards, 

Reviewer 4 Report

Comments and Suggestions for Authors

The author improve well.